

# Epidemiology and management of Fusarium wilt of *Eucalyptus camaldulensis* through systemic acquired resistance

Irfan Ahmad[1,*], Komal Mazhar[1,*], Muhammad Atiq[2],
Amna Kahtan Khalaf[3], Muhammad Haroon U. Rashid[1],
Muhammad Asif[1], Salman Ahmed[4], Zoha Adil[2], Amna Fayyaz[5],
Mohammad Khalid Al-Sadoon[6] and Hamad S. Al-Otaibi[6]

[1] Department of Forestry and Range Management, University of Agriculture Faisalabad, Faisalabad, Punjab, Pakistan
[2] Department of Plant Pathology, University of Agriculture Faisalabad, Faisalabad, Punjab, Pakistan
[3] College of Dentistry, Al-Bayan University, Baghdad, Iraq
[4] Department of Plant Pathology, University of Sargodha, Sargodha, Punjab, Pakistan
[5] Department of Plant Pathology, University of California, Davis, California, California, United States
[6] Department of Zoology, King Saud University, Riyadh, Saudi Arabia
* These authors contributed equally to this work.

Corresponding authors
Irfan Ahmad,
irfanahmad@uaf.edu.pk
Muhammad Haroon U. Rashid,
haroon.r@uaf.edu.pk

## ABSTRACT

*Eucalyptus camaldulensis* is a multifunctional tree and is globally used for the reclamation of problematic lands. *Eucalyptus camaldulensis* is prone to attack by a number of pathogens, but the most important threat is the Fusarium wilt (*Fusarium oxysporum*). Keeping in view the importance of *E. camaldulensis* and to manage this disease, five plant activators, *i.e.*, salicylic acid ($C_7H_6O_3$), benzoic acid ($C_7H_6O_2$), citric acid ($C_6H_8O_7$), dipotassium phosphate ($K_2HPO_4$), monopotassium phosphate ($KH_2PO_4$) and nutritional mixture namely Compound (NPK) and nutriotop (Fe, Zn, Cu, B, Mn) were evaluated in the Fusarium infested field under RCBD in the Research Area, Department of Forestry and Range Management, University of Agriculture, Faisalabad (UAF). Among plant activators, salicylic acid and a combination of compound + nutriotop exhibited the lowest disease incidence and enhanced fresh and dry weight of leaves compared to other treatments and control. Results of the environmental study indicated maximum disease incidence between 35–40 °C (max. T), 6–25 °C (mini. T), 70–80% relative humidity and 1.5–2.5 km/h wind speed while pan evaporation expressed weak correlation with disease development. It was concluded that Fusarium wilt of *Eucalyptus camaldulensis* could be managed through activation of the basal defense system of the host plant with provision of salicylic acid and balanced nutrition by considering environmental factors. Recent exploration is expected to be helpful for future research efforts on epidemiology and ecologically sound intervention of Fusarium wilt of *Eucalyptus camaldulensis*.

## INTRODUCTION

*Eucalyptus camaldulensis* Dehnh. (Sufaida) belongs to the family Myrtaceae, found in tropical and subtropical regions. It is a fast-growing species, native to Australia that can grow under a wide range of climate and edaphic conditions and also known as river red gum (*Minhas, Yadav & Bali, 2020*; *Dawar et al., 2007*; *Ferreira, Boyero & Calvo, 2019*). It is native to Australia and is known as river red gum. Pakistan, India, Algeria, East Africa, Sudan, Malaysia, Philippines, and Ethiopia are Eucalyptus growing countries of the world, and according to an estimate, its plantation has reached more than 20 million hectares (*Jagger & Pender, 2003*). Globally, Eucalyptus is used in agroforestry for reclamation of problematic soils (*Joshi, 2011*). Commonly grown regions of Eucalyptus are Sindh and Punjab in Pakistan. This multipurpose tree is a rich source of byproducts like honey, medicinal extracts and perfumes. Essential oils extracted from Eucalyptus leaves have shown a remarkable antiseptic activity against a wide range of infectious bacteria, viruses, and fungi (*Inouye, Takizawa & Yamaguchi, 2001*). Eucalyptus is prone to be attacked by a number of fungal, bacterial, and viral pathogens in different regions, but *Fusarium oxysporum* is responsible for Fusarium wilt disease poses significant threat (*Salerno, Gianinazzi & Gianinazzi-Pearson, 2000*). *F. oxysporum* is a soil-born pathogen that attacks roots and colonizes vascular tissue, causing wilts in Eucalyptus. Chlorosis, wilting, necrosis, immature leaf dropping, stunting, damping off, and browning of the vascular system are the characteristic symptoms of Fusarium wilt of Eucalyptus (*Salerno, Gianinazzi & Gianinazzi-Pearson, 2000*; *Iori, 2002*). The first disease symptom can be observed on the plants in the form of pale green streaks on the base of the petiole. *F. oxysporum* is a soil-born pathogen that attacks roots and colonizes vascular tissue, causing wilts in Eucalyptus (*Iori, 2002*).

The pathogen population and their attack on the host plant are interlinked with the environment (*Misra et al., 2020*). Thus, it is necessary to study the effect of all the environmental factors involved in the development of the pathogen's population dynamics and disease development (*Misra et al., 2020*; *Saremi, Burgess & Backhouse, 1999*). It is of prime importance for the management of Fusarium wilt of eucalyptus, as climate has a major role in the distribution of Fusarium species which are thermophilic in nature and are significantly correlated with soil and environmental factors (*Saremi, Burgess & Backhouse, 1999*). High temperature and moisture are favorable for the disease occurrence. Reduction in soil moisture is a significant strategy for managing this disease (*Madhavi & Bhattiprolu, 2011*). *Land, Cortes & Diaz (2006)* and *Larkin & Fravel (2002)* reported that 17–24 °C air and 15–25 °C soil temperature played a crucial role in developing Fusarium wilt disease. Therefore, sudden fluctuations in environmental conditions can increase wilt disease incidence. In various regions researchers are now compelled to investigate the role of biotic and abiotic factors facilitating disease emergence.

In the light of the above-mentioned facts the study investigated how soil and ambient environmental conditions affect the growth of Eucalyptus Fusarium wilt.

Different control measures like cultural practices, chemical, and biological approaches are in practice against Fusarium wilt (*Kamal, Elyousr & Hashem, 2009*) but activation resistance through resistant cultivars is likely the most appropriate one (*Nelson et al., 2018*). If the disease appears in epidemic form, then farmers mostly use chemicals due to their quick response and easy availability, but excessive use chemicals cause environmental pollution which affects human health due to their residual effects. On the other hand, continuous use of chemicals causes resistance in pathogens, which may lead to the development of more virulent strains of pathogen due to mutation. This issue forces scientists, researchers, and farmers to search for some alternatives to chemicals. In this scenario, In this scenario, use of plant defense activators is likely the best strategy for plant disease management (*Singh et al., 2020*; *Koo, Heo & Choi, 2020*) because plant defense activators have the least deleterious effects on the environment, humans, and plants (*Huang & Hsu, 2003*). They activate the plant defense system through the production of phenolic antioxidants in the plant (*Tuladhar, Sasidharan & Saudagar, 2021*). So, in the current study, different plant defense activators at different concentrations were evaluated towards the Fusarium wilt of eucalyptus.

The use of macro and micro nutrients is another strategy to manage the Fusarium wilt of eucalyptus (*Asma et al., 2009*). Deficiency or excessive use of nutrients is responsible for causing ill effects on the plants by causing a disturbance in the metabolism and physiology of the host plants, which makes the plants vulnerable to pathogens (*Asma et al., 2009*; *Stenger et al., 2021*). The deficiency of nutrients in plants is the prime cause of the deteriorating defense system of the plants, which makes the plants susceptible to various diseases (*Shrestha et al., 2020*). The application of a balanced amount of nutrients acts as the first line of defense against invading pathogens by activating the cross-protection mechanism of the host plant through production of different biochemical enzymes and phenolic antioxidants (*Saleem et al., 2021*).

Due to the detrimental effects of the mentioned disease, it is the need of hour to investigate the role of nutrients, plant defense activators and climatic factors in disease management.

## MATERIALS AND METHODS

### Identification and purification of pathogen

Plant parts showing typical symptoms of the disease were collected from Shorkot (Punjab) Irrigated Forest Plantation using the rectangular survey method. The principal species of this plantation (*E. camaldulensis*) suffered from various diseases, but the most dominant one was Fusarium wilt. Five diseased samples from 10 different sites were collected and brought to the Plant Pathology laboratory to isolate pathogens associated with Eucalyptus's wilt. Initially, infected roots were washed with distilled water and then surfaced sterilized (1% Sodium hypochlorite) followed by two washings of distilled water. Then samples were cut into small pieces (0.5–1 cm). Pieces of root were dried on a watch

glass with the help of sterilized filter paper and were placed in sterilized Petri plates containing potato dextrose agar medium (PDA) (Water = 1,000 mL, Potato = 200 g, Dextrose = 20 g, Agar = 20 g autoclaved at 121 °C and 15 PSI for 15 min) and 48–72 h of incubation at 30 at °C to promote fungal development (*Pempee et al., 2020*). The pathogen was then purified by transferring the fungal colony into another Petri plate containing PDA. After 5 days, the pathogen was identified based on morphological and taxonomical characteristics (colony color, pattern of fungal growth, shape, and size of spore) through microscopy and pathogenicity test (*Soesanto, Utami & Rahayuniati, 2011*).

## Pathogenicity test

A pathogenicity test of *F. oxysporum* inoculum was conducted to fulfill Koch's postulates. For this purpose, seedlings of *E. camaldulensis* (10–16 cm height) were collected from the nursery of the Department of Forestry and Range Management, UAF and transferred into pots (30 cm diameter) containing sandy loam soil, @ two seedlings/ pot under completely randomized design (CRD) and were kept under strict and careful observation in the greenhouse, situated in the Research Area, Department of Forestry and Range Management. Twelve plants of 60 days' age were used for the pathogenicity test. Among these plants, nine plants were treated with inoculum, while three plants were only drenched and sprayed with distilled water. Plants were inoculated through the soil drenching and spraying method (*Paulino et al., 2020*) by using $1 \times 10^6$ spores/mL of water, measured with the help of a hemocytometer (ZNC-30). Symptoms appeared after 10 days of inoculation. Then, the pathogen was re-isolated from artificially inoculated eucalyptus plants to fulfill Koch's postulates (*Haq, Ijaz & Khan, 2022*).

## Assessment of nutrients against Fusarium wilt of *E. camaldulensis* under greenhouse conditions

Earthen pots (30 cm) were filled with homogeneous sterilized soil with a 1:1:1 ratio (sand, silt and farmyard manure) and six-month-old seedlings of *E. camaldulensis* were transferred into these earthen pots. Seedlings were arranged under Completely Randomized Design (CRD) with five replications under greenhouse conditions, situated in the Research Area Department of Forestry and Range Management. Seedlings were inoculated (only one time) with a spore suspension of *Fusarium oxysporum* @$1 \times 10^6$ spores/mL of $H_2O$, which was adjusted by using a hemocytometer (ZNC-30) through soil drenching and spraying method. Four treatments namely $T_1$ (compound ($N = 20\%$, $P = 20\%$ and $K = 20\%$)), $T_2$ (Nutriotop (Fe = 30%, B = 10%, Cu = 10%, Zn = 40%, Mn = 10%)) and $T_3$ (combination of $T_1 + T_2$) while fourth treatment ($T_4$) was distilled water only. All the nutrients were applied in liquid form under controlled conditions to induce resistance towards Fusarium wilt of *E. camaldulensis* under greenhouse conditions (FW). T1 @ 5 g, 7.5 g, 10 g/L, $T_2$@ 0.25, 0.5, 0.75 g/L and combination of $T_1 + T_2 = (T_3)$ along with control ($T_4$) were evaluated towards wilt of Eucalyptus. Data regarding incidence was noted throughout the season with one-week intervals (*Ashfaq et al., 2014*) by using the following formula.

Disease incidence (%) = ((No. of infected plants)/(No. of total plants)) × 100

## Establishment of Fusarium infested field in the research area department of forestry and range management

Fusarium infested field was established for the evaluation of different nutrients and plant defense activators as a management approach toward wilt. For this purpose, an area of 400 m$^2$ was selected in the Research Area, Department of Forestry and Range Management, which was continuously cultivated with Eucalyptus seedlings for the last 5 years. Two sprays of inoculum were done on the selected area consecutively to enhance the population pressure of Fusarium in the soil when assessing the efficacy of plant defense activators and nutrients towards Fusarium wilt of eucalyptus. Seedlings of eucalyptus were planted in the field and again infused with spore suspension @ $1 \times 10^6$ spores/mL of H$_2$O. This spore concentration was measured by using a hemocytometer. The spores in this solution were made by mixing 3–4 mL of distilled water in a Petri plate that has a 7–10-days old culture of inoculum. It was poured into a beaker containing 1,000 mL of water and shaken well and drenched near the seedlings' root zone to enhance the amount of inoculum for disease establishment.

## Assessment of plant activators against Fusarium wilt of *E. camaldulensis* under field conditions

Five plant activators, *i.e.*, salicylic acid (C$_7$H$_6$O$_3$), benzoic acid (C$_7$H$_6$O$_2$), citric acid (C$_6$H$_8$O$_7$), dipotassium phosphate (K$_2$HPO$_4$), monopotassium phosphate (KH$_2$PO$_4$) along with control @ 0.25, 0.75 and 1% with three replications were applied against Fusarium wilt. For this purpose, surface sterilized seeds of Eucalyptus with a 1% solution of sodium hypochlorite were sown under field conditions attributed by Randomized complete block design (RCBD) in the sick field. When the seedlings were six months old, all the plant activators at three different concentrations were applied to create resistance in Eucalyptus plants through soil drenching and spray methods. To enhance inoculum pressure, an amount of $1 \times 10^6$ spores/ mL of H$_2$O was applied to the plants for the development of wilt disease. Upon appearance of disease symptoms, data regarding disease incidence was recorded by using the following formula with one-week intervals throughout the season.

Disease incidence (%) = (No. of infected plants)/(No. of total plants) $\times$ 100

## Impact of plant activators and nutrients on the number of leaves, fresh and dry weight

Data regarding no. of leaves and fresh and dry weight of leaves (g) was recorded with the help of electric balance (AEL-223) after the application of plant activators (*i.e.*, Salicylic acid, benzoic acid, citric acid, dipotassium phosphate, monopotassium phosphate). These activators along with control at three different concentrations were evaluated @ 0.25%, 0.75% and 1% and nutrients (Compound and Nutriotop) alone and in combination at three concentrations while control plants were treated only with distilled water. Data regarding the increase in the number of leaves and their fresh and dry weight were recorded at the end of the season. Data regarding number of leaves fresh and dry weight were recorded after 10, 20, and 30 days on intervals.

## Characterization of climatic factors favoring Fusarium wilt of *Eucalyptus camaldulensis*

To determine the impact of epidemiological factors on the development of *Fusarium wilt*, seedling of *Eucalyptus camaldulensis* were transplanted in the Research Area, Department of Forestry and Range Management under natural field conditions and data regarding disease incidence was recorded for the whole season. No artificial inoculation was done on seedlings as these were planted in an already infested field. Data regarding environmental factors like maximum and minimum temperature (°C), wind speed (Km/h), rainfall (mm), relative humidity (%) and pan evaporation (%) was collected from the metrological observatory situated at the Research Area, Department of Agronomy, University of Agriculture, Faisalabad. The influence of these environmental factors was observed on the disease development through correlation (*Steel, 1997*). Regression analysis identified the most conducive environments for disease progression.

## Data analysis

All statistical tests regarding the impact of epidemiological factors like temperature (max. and mini), relative humidity, rainfall, wind speed, pan evaporation, different nutrients, and plant defense activators on Fusarium wilt of *Eucalyptus camadulensis* were performed by using SAS/STAT statistical software (SAS Institute, Cary, NC, USA). A greenhouse experiment was conducted under CRD, while field experiments were conducted under RCBD. Means were separated using Fisher's protected least significant difference (LSD) procedure by taking a 5% probability level (*Steel, 1997*). Analysis of variance (ANOVA), the interaction of different treatments and their combinations were developed by using SAS/STAT software package (SAS Institute, Cary, NC, USA).

# RESULTS

## Symptomology and development of Fusarium wilt of *E. camaldulensis* under field conditions

Younger leaves showed vein clearing, marginal tissue necrosis, and yellowing, whereas older plants showed withering of leaves followed by falling and eventually mortality. The soil-borne disease invaded the plants *via* vascular systems. The disease started to appear during the last week of April and reached its peak during June-July.

## Effect of compound and nutriotop at different concentrations towards wilt, dry wight, fresh weight and no. of leaves of *E. camaldulensis* under field conditions

Nutrients were evaluated alone and in combination against the fusarium wilt of Eucalyptus. Minimum disease incidence was exhibited by nutrients provision in combination, *i.e.*, compound + nutriotop (9.567%) followed by compound (16.88%) and nutriotop (16.88%) as compared to control. In the case of interaction between treatments and their concentrations, minimum disease incidence was expressed by nutrients provision in integration, *i.e.*, compound + nutriotop (10.20%, 9.40%, 9.10%) followed by

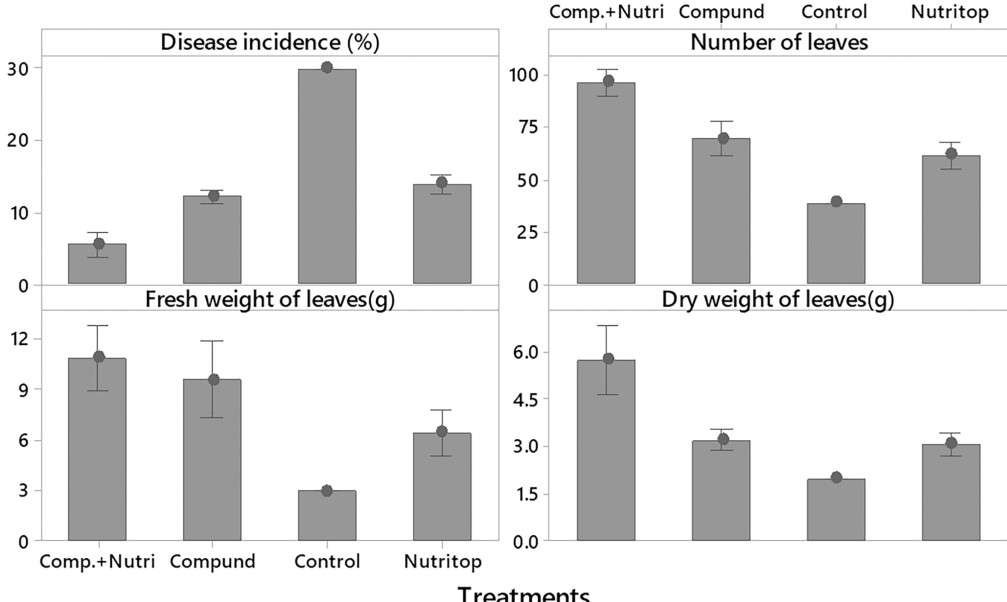

*Individual standard deviations are used to calculate the intervals.*

**Figure 1 Impact of treatments (nutrients) on the incidence of Fusarium wilt, number of leaves (NL) and fresh weight (FW) of leaves under field conditions.** Note: compound ($C_1$ = 4,000, $C_2$ = 5,000, $C_3$ = 6,000 mL/ha); Nutriotop ($C_1$ = 1,000, $C_2$ = 1,200, $C_3$ = 1,400 Ml/ha); Compound + Nutirtop ($C_1$ = 3,000, $C_2$ = 4,000, $C_3$ = 5,000 mL/ha).

compound (20%, 17%, 14%), nutriotop (22%, 19%, 14%) as compared to control (30.20%) (Fig. 1). Maximum dry weight of leaves was observed when nutrients were applied in combination (4.677 g) followed by nutriotop (3.167 g) and compound (3.012 g) as compared to control. While in case of interaction b/w treatment and concentration combination of compound + nutriotop exhibited maximum dry weight followed by nutriotop and compound as compared to control, similar observations were noted in the case of fresh weight and number of leaves after application of nutrients alone, in combination and interaction between treatments and concentrations (Fig. 2).

## Effect of plant activators on fusarium wilt occurrence, fresh, dry weight, and number of leaves of *E. camaldulensis* under field conditions

Among all plant activators, the minimum incidence of fusarium wilt was 11.383% when Salicylic acid was used, followed by dipotassium hydrogen phosphate, potassium dihydrogen phosphate, benzoic acid, and citric acid (15.11%, 17.556%, 17.773% and 18. 567%) respectively as compared to control (Fig. 3). While in the case of interaction b/w treatments and concentrations maximum incidence of disease was expressed by benzoic acid (24.567%, 16%,13%) @ 0.5%, 0.75% and 1%, respectively, while minimum disease incidence was exhibited by salicylic acid (13.66%, 12.33% and 8%) at three concentrations respectively (Fig. 4) as compared to control. The maximum no. of leaves, fresh and dry weight of leaves expressed by salicylic acid, followed by dipotassium hydrogen phosphate,

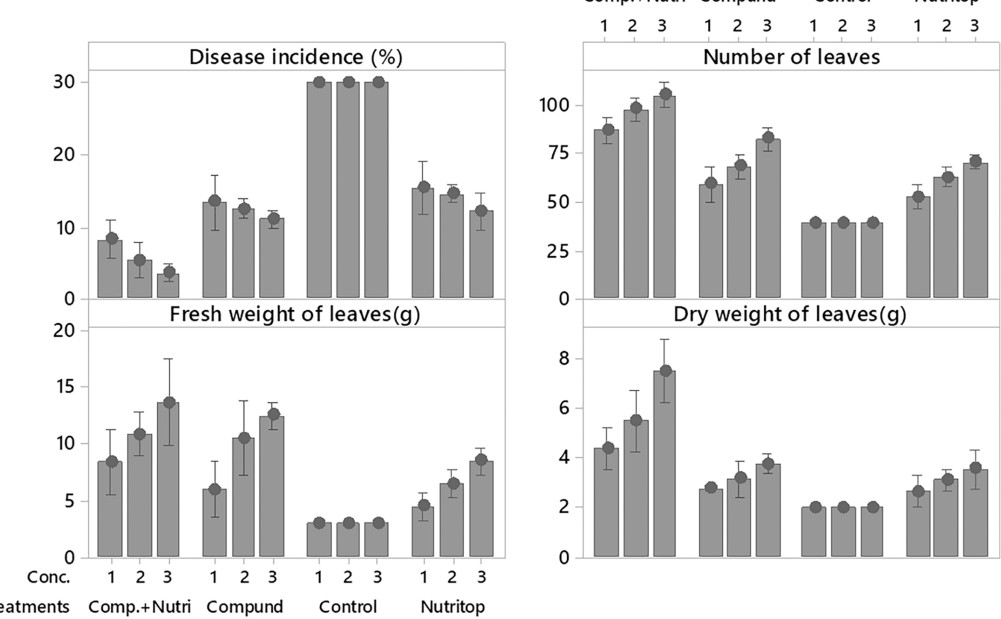

**Figure 2 Evaluation of different nutrients and their concentration on the incidence of Fusarium wilt, number of leaves (NL) fresh and dry weight of *E. camaldulensis* leaves under field conditions.** Note: Compound = (N = 20%, P = 20% and K = 20%), Nutriotop = (Fe = 30%, B = 10%, Cu = 10%, Zn = 40%, Mn = 10%), LSD: Disease incidence (Treatments = 0.3429, Treatments × Concentration = 0.5939), NL (Treatments = 2.5768, Treatments × Concentration = 6.4491), FW (Treatments = 0.4774, Treatments × Concentration = 0.8269). Note: Compound (C1 = 4,000, C2 = 5,000, C3 = 6,000 mL/ha); Nutriotop (C1 = 1,000, C2 = 1,200, C3 = 1,400 Ml/ha); Compound + Nutirtop (C1 = 3,000, C2 = 4,000, C3 = 5,000 mL/ha).

potassium dihydrogen phosphate, benzoic acid, and citric acid alone and in interaction with all concentrations, respectively (Fig. 4) as compared to control.

## Characterization of environmental conditions conducive for the development of Fusarium wilt of *E. camaldulensis* under field conditions

Disease symptoms started to appear at 25 °C and it continuously increased up to 42 °C (maximum temperature) as indicated by $R^2 = 0.6488$. Maximum disease incidence was observed b/w 35–40 °C. The regression equation indicated that with increase in one-unit temperature, the disease increased by 1.373 units. A significant positive relationship was noted in the case of minimum temperature ($R^2 = 0.5694$). Increased in disease incidence was noted b/w 6–28 °C, and maximum disease was noted b/w 6–12 °C under minimum temperature. Relative humidity and wind speed likewise expressed positive relation with the Fusarium wilt with $R^2 = 0.8258$ and 0.5867, respectively. Progress in disease was noted b/w 60%–90% relative humidity and 1–3.5 mm rainfall. Maximum disease in case of relative humidity was observed b/w 70%–80% and 1.5–2.5 km/h wind speed. Pan evaporation expressed a very weak relationship with disease development, as shown in

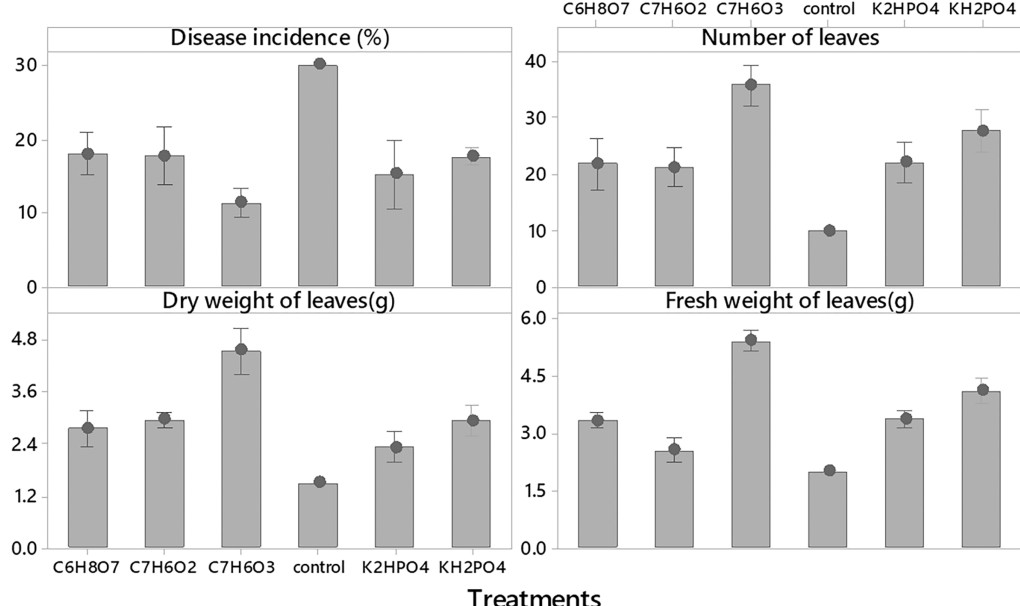

*Individual standard deviations are used to calculate the intervals.*

**Figure 3** Evaluation of different plant activators on the incidence of Fusarium wilt, number of leaves (NL) fresh and dry weight of *E. camaldulensis* leaves under field conditions.

regression models (Fig. 5) and scattered plots of maximum, and minimum temperature, relative humidity, rainfall, wind speed and pan evaporation (Fig. 6). Maximum disease was noted during the first week of September (Fig. 7).

## Evaluation of nutrients at different concentrations towards wilt, dry wight, fresh weight and no. of leaves of *E. camaldulensis* under greenhouse condition

Nutrients were evaluated alone and in combination against the fusarium wilt of Eucalyptus. Minimum disease incidence was exhibited by nutrients provision in combination, *i.e.*, compound + nutriotop followed by compound and nutriotop as compared to control (Fig. 8). In the case of interaction between treatments and their concentrations, minimum disease incidence was expressed by nutrients provision in integration, *i.e.*, compound + nutriotop followed by compound, nutriotop asnutriotop as compared to control (Figs. 8 and 9). Maximum dry weight of leaves was observed when nutrients were applied in combination followed by nutriotop and compound as compared to control. In the case of interaction b/w treatment and concentration combination of compound + nutriotop exhibited maximum dry weight followed by nutriotop and compound as compared to control, similar observations were noted in the case of fresh weight and number of leaves after application of nutrients alone, in combination and interaction between treatments and concentrations (Fig. 9). A Fisher's least significant difference (LSD) 0.05 was calculated to determine differences among the treatment means ($P < 0.05$) for each variable.

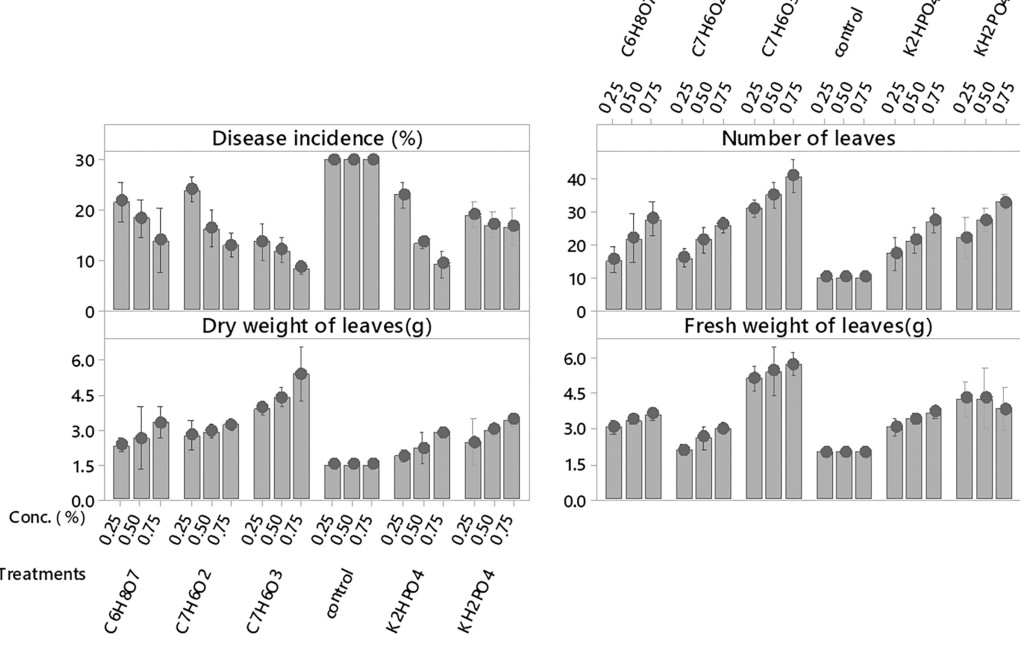

Figure 4 Evaluation of different plant activators and their concentration on the incidence of Fusarium wilt, number of leaves (NL) fresh and dry weight of *E. camaldulensis* leaves under field conditions. Note: All the plant activators were applied @ three concentrations (0.25%, 0.75% and 1%, LSD): Disease incidence (Treatments = 0.5089, T × C = 0.8815), NL (Treatments = 1.1176, Treatments × Concentration = 1.9357), FW (Treatments = 0.1571, T × C = 0.2722). Salicylic acid ($C_7H_6O_3$), Benzoic Acid ($C_7H_6O_2$), Di-potassium hydrogen phosphate ($K_2HPO_4$), potassium di-hydrogen phosphate ($KH_2PO_4$), Citric Acid ($C_6H_8O_7$).

## DISCUSSION

Fusarium wilt is a major issue in Eucalyptus growing areas in Pakistan. Main characteristic symptoms of this disease are vascular discoloration, chlorosis, and wilting (*Alegbejo et al., 2006*). There are many factors that affect disease development such as inoculum density, population, host range and time of infection (*Rekah, Shtienberg & Katan, 2010*). Temperature, pH, carbon, and nitrogen availability also influence the dissemination of disease symptoms (*Naik, Rani & Madhukar, 2008*). Similarly, several other factors like environment, infectious pathogen and host range are involved in the occurrence of disease (*Saremi & Amiri, 2010*). The most effective and practical strategy for lowering the prevalence of soil-borne diseases is the use of resistance sources (*Saremi & Amiri, 2010*; *Naik, Madhukar & Rani, 2007*). Furthermore, the use of a resistance source decreases the incidence of Eucalyptus wilt and avoids fungicide toxicity (*Naik, Madhukar & Rani, 2007*).

### Evaluation of nutrition towards incidence of fusarium wilt, number of leaves along with fresh and dry weight

Physiological functions of plants are destroyed by the effect of pathogen, such as movement of important nutrients, their absorption, translocation, and utilization from plant roots towards shoot system (*Stewart et al., 2005*; *Khan et al., 2014*) as nutrients play a vital role in plants. The most important purpose of nutrients is to maintain plant health by

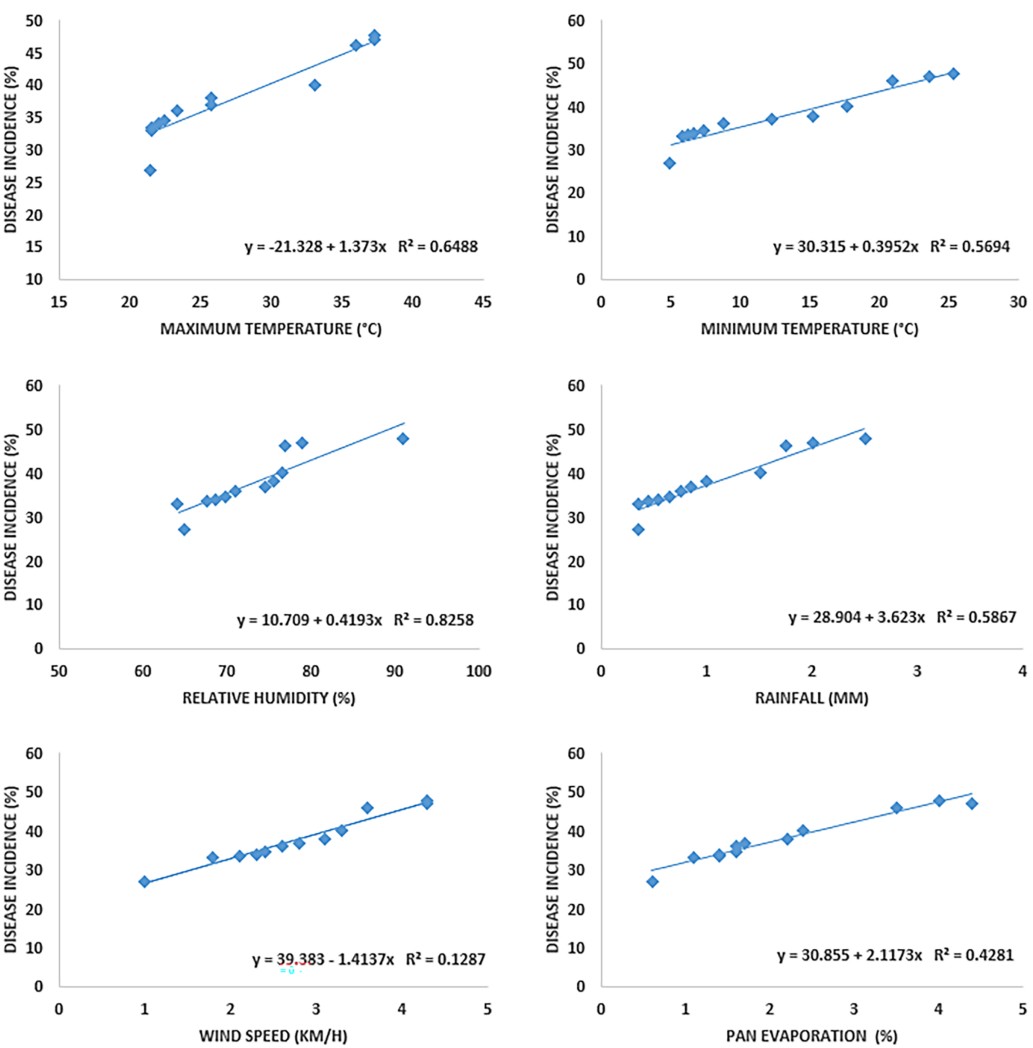

**Figure 5 Relationship b/w maximum, minimum temperature (°C), relative humidity (%), rainfall (mm), wind speed (km/h) and pan evaporation with the development of Fusarium wilt of *E. camaldulensis*.**

regulating different cellular functions through the activation of metabolism, making plants more resistant to pathogen attack (*Saikia et al., 2009*). Therefore, the suitable accessibility of micro (Zn, Mn and Fe, Cu and B) and macronutrients (NPK) decrease the incidence of plant diseases because nutrients structurally stabilize the protein molecule, whereas majority of them are catalytically active co-factor in enzymes (*Allabi, 2006*; *Suharaja & Sutarno, 2009*). Nutrient availability in plants is reduced due to the attack of pathogen, so this nutrient deficiency deteriorates the plant's growth, development, and defense system (*Sanjeev & Eswaran, 2008*). In the present study, nutriotop (Fe, Cu, Zn, B and Mn) solo and in grouping with the composite mixture (NPK) was assessed against Fusarium wilt disease of Eucalyptus. The results demonstrated that compounds with nutriotop amplified resistance in plants, ultimately decreasing the incidence of Fusarium wilt.

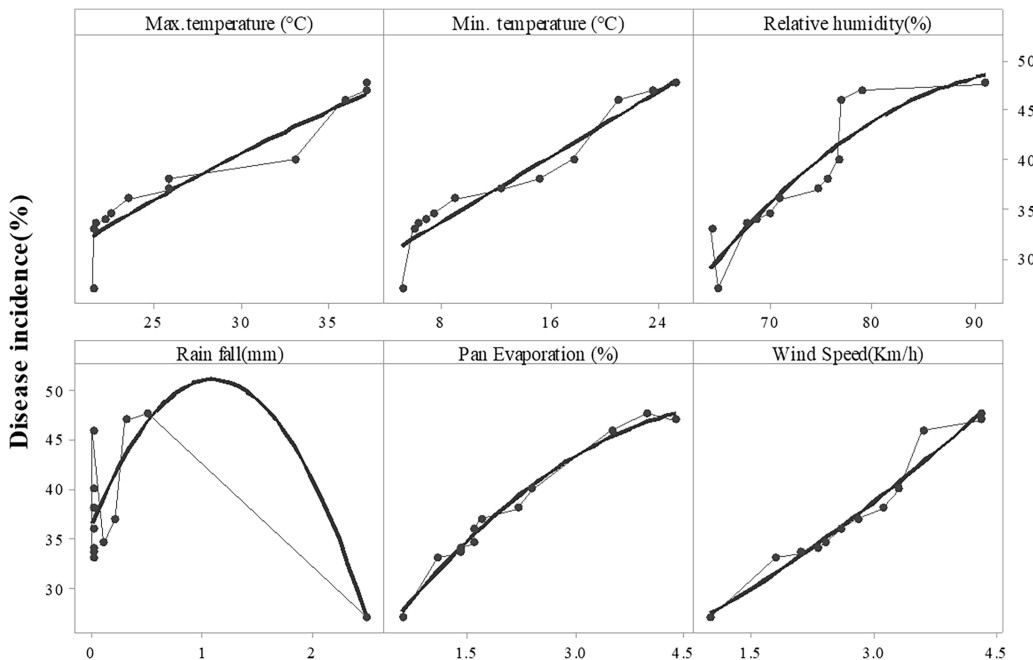

**Figure 6 Relationship b/w monthly maximum, minimum temperature (°C), relative humidity (%), rainfall (mm), wind speed (km/h) and pan evaporation with the development of Fusarium wilt of *E. camaldulensis*.** Note: Upper line showed the disease incidence and lower line was showed max. mini. Temperature, rainfall, relative humidity and wind speed.

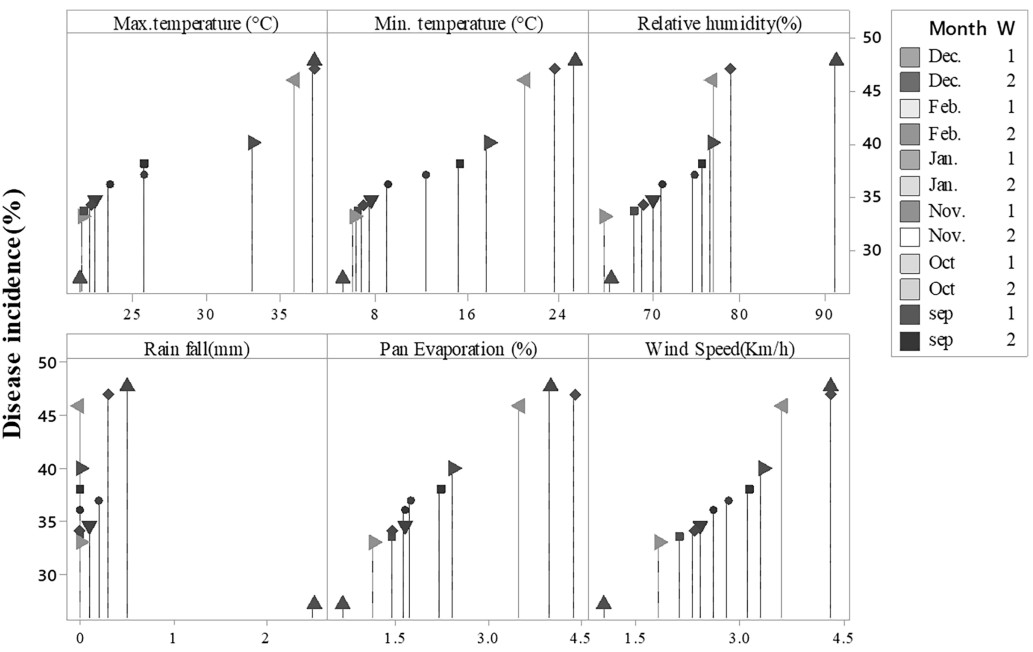

**Figure 7 Incidence of Fusarium wilt of *E. camaldulensis* during different months.**

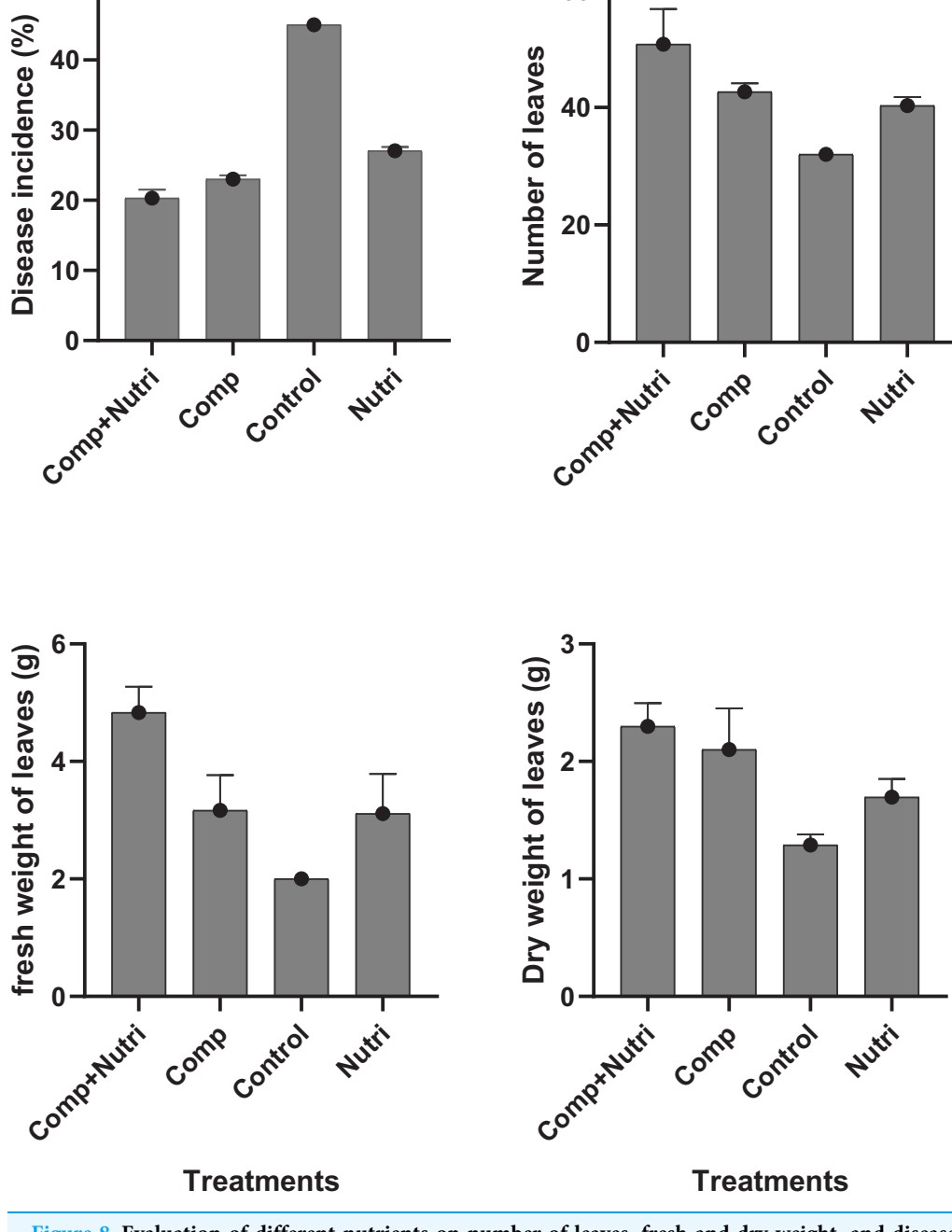

**Figure 8 Evaluation of different nutrients on number of leaves, fresh and dry weight, and disease incidence against fusarium wilt of Eucalyptus under greenhouse condition.**

Nitrogen is the most important part of plant nutrients as it is an integral part of different molecules such as protein, amino acid, nucleic acid, chlorophyll cytosine and auxin (*Guertal, 2000*), plants fulfill their nitrogen deficiency from nitrate and ammonium ions (soil). When a pathogen attacks Eucalyptus seedlings, it snatches nitrogen from the plant. Consequently, plant growth is inhibited by the attack of *Fusarium oxysporum* (*Chellemi & Lazarovits, 2002*). Similarly, some other nutrients like zinc (Zn), boron (B), and

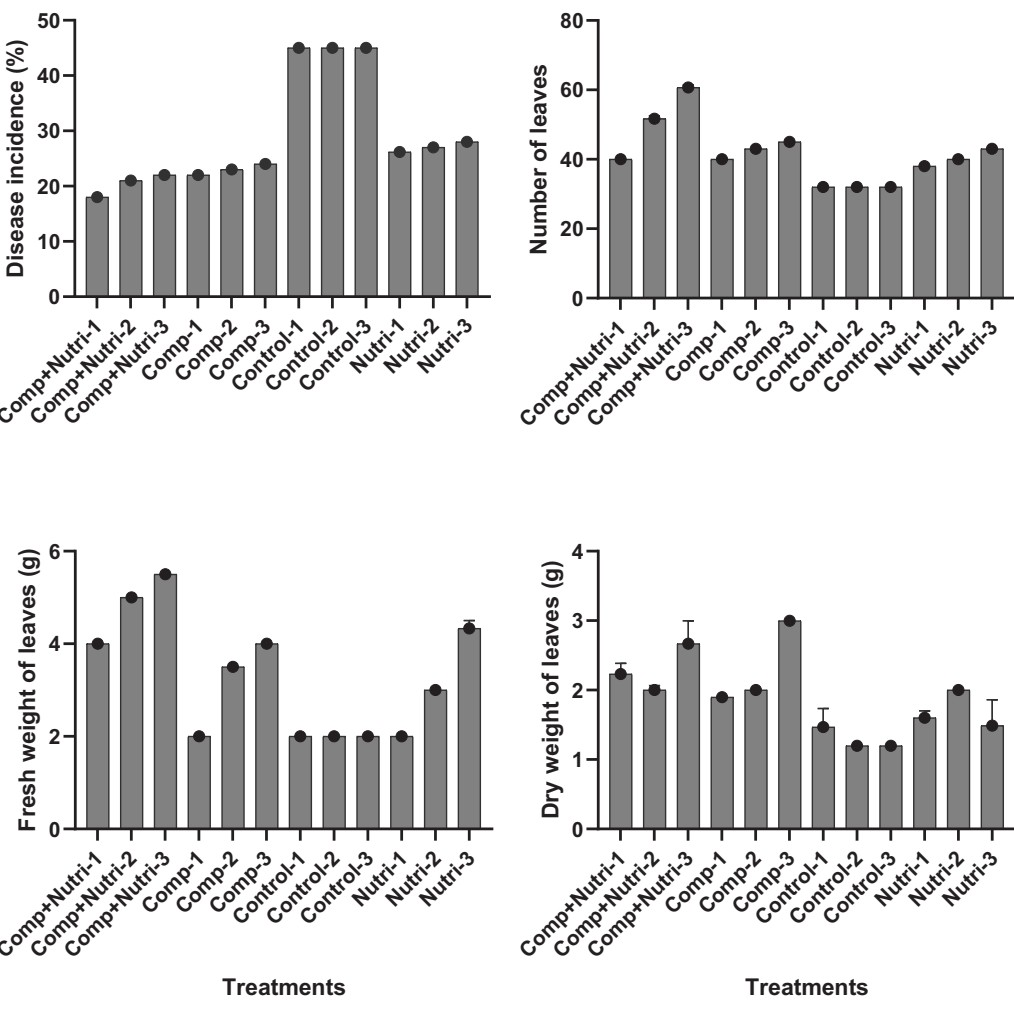

**Figure 9 Evaluation of different nutrients and their concentrations on number of leaves, fresh and dry weight, and disease incidence against fusarium wilt of Eucalyptus under greenhouse condition.**

phosphorus (P) have a critical role in maintaining plant health. Phosphorus is an important component of ATP, DNA, RNA, and cell membrane (*Suharaja & Sutarno, 2009*). Zinc is also a significant element that performs a fundamental responsibility in the production of nitrogen and carbon-based components. Sugar levels and amino acid pro-duction in plant tissue are reduced by the low level of zinc. It also detoxifies superoxide radicals, which protects the cell membrane from oxidative damage (*Cakmak, 2000*). Critical damage to the cell membrane caused by free radicals can collapse the cell membrane, ultimately facilitating food delivery to the pathogen (*Mengel & Kirkby, 2001*).

In the current study, the application of zinc reduced disease and increased the amount of foliage and their dry and fresh weights, which is supported by the work of *Grewal, Graham & Rengel (1996)*. Current investigation was in line with previous study in which the rate of nitrogen application significantly impacted on plant biomass and Fusarium wilt severity (*Orr et al., 2022*). Recent findings were advocated by previous study in which minimum disease incidence was observed by the application of micronutrients in
combination and significant increase in plant biomass was also noticed (*Abd-El-Rahman, Mazen & Khalil, 2014*).

## Impact of plant activators on Fusarium wilt of Eucalyptus, number of leaves along with their fresh and dry weights

The defense systems of plants become inactivated by the attack of pathogens, and the chance of disease enhancement increases. For the initiation of defense system in plants, resistance genes play crucial role towards pathogen. Use of growth regulators activate the basal defense system of the host plant (*Jalali, Bhargava & Kamble, 2006*; *Inoue, Namiki & Tsuge, 2002*), and susceptible plant become resistant (*Elwan & EL-Hamahmy, 2009*). Accumulation of indole acetic acid and salicylic acid enhanced the growth and resistance of plants (*Hayat et al., 2005*). Moreover, the action of resistance genes was increased by the adequate quantity of salicylic acid (*Khan, Prithviraj & Smith, 2003*).

In the present study, the role of five different plant activators, Salicylic acid ($C_7H_6O_3$), Benzoic acid ($C_7H_6O_3$), Citric acid ($C_6H_8O_7$), Dipotassium phosphate ($K_2HPO_4$), Monopotassium phosphate ($KH_2PO_4$) was investigated towards the occurrence of Fusarium wilt in Eucalyptus trees and least disease incidence was expressed by applying salicylic acid. The recent study agreed with (*Hanieh et al., 2013*; *Ali, Smith & Guest, 2000*; *Pempee et al., 2020*), who estimated different growth regulators against Fusarium wilt, *i.e.*, Salicylic acid, Ferric chloride, Hydrogen peroxide, Dipotassium hydrogen orthophosphate, Indole acetic acid and Calcium chloride, but Salicylic acid revealed prominent findings in response to Fusarium wilt. Therefore, by the activation of reactive oxygen species (ROS) and fluctuation in cell wall components, salicylic acid is activated in plant defense through this process (*Colson-Hanks, Allen & Deverall, 2000*; *Agrios, 2005*; *Govindappa, Rai & Lokesh, 2011*), and it increases the production of various proteins (*Hayat, Ali & Ahmad, 2007*; *EL-Yazied, 2011*; *Biswas, Paandey & Rajik, 2012*). Hence, plants expressed minimum disease with a maximum number of leaves and maximum amount of fresh and dry weights, as observed in the contemporary study. Salicylic acid-induced production of Hydrogen peroxide ($H_2O_2$) is a vital part of ROS. Hydrogen peroxide retarded the fungal growth due to antifungal activity and contributed to the amplification of phenoxy-radicals in plant cells (*Inbar et al., 1998*; *Huynh et al., 1996*). Although the breakdown of the membrane is started by Lipid peroxidation (*Lee, Leslie & Bowden, 2006*). Conversely, antioxidant defense is a major immune system in plants that protects them from oxidative stress and suppresses the partial production of ROS (*Sitara & Hasan, 2011*; *Torres-Castillo et al., 2013*).

## Impact of climatic factors on the incidence of Fusarium wilt of *E. camaldulensis*

Climate change is an issue of global concern (*Bhatti et al., 2006*). Climate change has the potential to initiate positive and negative interacting processes which affect forests (*Williams et al., 2010*). For example, maximum concentrations of $CO_2$ in the atmosphere increase the tree's growth rate and water use efficiency (*Boisvenue & Running, 2010*), and it influences the host-pathogen interactions. Hence, this changing climate also impacted the

behavior and distribution of pathogens. Some pathogens (including fungi and bacteria), directly and indirectly, are affected by climate. Under favorable environmental conditions, the pathogen's life cycle is directly affected by temperature and moisture because most pathogens are sensitive to precipitation and moisture. While the rate of reproduction, spreading, and infection are higher when conditions become conducive for them. Similarly, when plants are under stress, pathogens are indirectly affected by environmental factors to their host plants (*Boisvenue & Running, 2010*; *Van Mantgem et al., 2009*).

Meanwhile, it's very difficult to judge whether climate alone causes tree mortality, as the number of reports regarding dieback, decline and mortality attributed to climate drivers is increasing. If a tree died due to heat and drought, this phenomenon is associated with climate change, but irregular droughts have long been involved in mortality (*Van Mantgem et al., 2009*). If an increase in mortality occurred in all trees, then it may be due to climate change. Some forested ecosystems of the world may already be affected by climate change, and a further rise in mortality rate is assumed to be linked to a scarcity of water (*Allen et al., 2010*). Changing atmospheric composition and climate can also modify the plant canopy. Elevated $CO_2$ fertilization stimulates plant growth morphology, physiology, chemical composition and gene expression (*Matros et al., 2006*). These morphological changes can include increased plant height, number of branches and tillers, thickness, and area of leaves.

The growth of plants is influenced by minimum soil and air temperature (°C), soil moisture (%), wind speed and (km/h) rainfall (mm) which are the vital causes that affect the plants for the infection growth (*Larkin & Fravel, 2002*; *Bakhsh, Iqbal & Haq, 2007*). Sudden changes in climatic factors can boost the resistance and susceptibility of host plant (*Chakaraborty & Pangga, 2004*; *Ahmad, Khan & Siddiqui, 2013*). As these environmental factors bring various changes in pathogens, such as reproduction, growth rate, host-pathogen interaction, infection, and dissemination (*Saremi, Burgess & Backhouse, 1999*; *Ghini, Hamada & Bettio, 2008*). Results of the current study showed a significant relationship between temperature (maximum and minimum), pan evaporation, wind speed and rainfall with disease development which is supported by the findings of Cherian and Varghese (*Cherian & Varghese, 2005*). These findings also agree with these studies (*Zakaria & Lockwood, 1980*; *Abawi & Barker, 1984*; *Khilare & Ahmed, 2012*; *Alegbejo et al., 2006*; *Agrios, 2005*), who reported that hot and humid weather positively impacted the prevalence of disease. A similar finding was also reported by *Westerlund, Campbell & Kimble (1974)*, *Karimi, Owuoche & Silim (2012)*, *Mehmood et al. (2013)*, *Land, Cortes & Diaz (2006)*. Increase in temperature amplified the multiplication rate of the pathogen (*Burgess et al., 2008*). This rapid multiplication of pathogens causes a direct effect on roots which results in the destruction of plants (*Emberger & Weltry, 1983*; *Senthilkumar, Madhanraj & Panneerselvam, 2011*). It was also observed that when the temperature was raised from 35 °C, the amount of $HNO_2$ and $NH_3$ was also raised after the disease attack (*Senthilkumar, Madhanraj & Panneerselvam, 2011*). Furthermore, the production of $HNO_2$ and $NH_3$ increased the pathogen's intensity and revitalized the pathogen's virulence (*Megie, Pearson & Mitbold, 1967*; *Benjumea et al., 2014*).

The results of the present study are helpful for Eucalyptus growers in managing fusarium wilt disease through the knowledge of conducive environmental variables. Because before contemporary study, no work was done on determining suitable environmental conditions for this disease. No doubt, the results of the current study are useful for farmers. Still, it is need of the hour to design such a type of study on mass scale to know the intensity and severity for developing the most reliable disease-predictive model and to validate the results under field conditions. It requires at least three to 5 years of data of disease and environmental factors.

## CONCLUSION

We can conclusively say that Fusarium wilt of *Eucalyptus camaldulensis* can be managed by plant activators, macro (NPK) and micronutrients (Fe, Zn, Cu, B, Mn) by keeping in view the environmental factors. Among all the treatments, salicylic acid and a combination of compound + nutriotop exhibited minimum disease incidence and the best growth response. Climate has a key role in the dissemination of Fusarium species, so the findings of the current study showed maximum disease incidence between 35–40 °C (max. T), 6–25 °C (mini. T), 70–80% relative humidity, 1.5–2.5 km/h wind speed while pan evaporation expressed week correlation with disease development.

### Future implications of recent revelations for growers

Growers' community can cope with Fusarium wilt of *Eucalyptus camaldulensis*, a formidable challenge to agroforestry by using currently investigated fruitful management tool.

### Funding

This work was supported by Researchers Supporting Project Number (RSP2024R410), King Saud University, Riyadh, Saudi Arabia. The funders had no role in study design, data collection and analysis, decision to publish, or preparation of the manuscript.

### Grant Disclosures

The following grant information was disclosed by the authors:
Researchers Supporting Project Number: RSP2024R410.
King Saud University.

### Competing Interests

The authors declare that they have no competing interests.

### Author Contributions

- Irfan Ahmad conceived and designed the experiments, performed the experiments, analyzed the data, prepared figures and/or tables, and approved the final draft.
- Komal Mazhar conceived and designed the experiments, performed the experiments, authored or reviewed drafts of the article, and approved the final draft.

- Muhammad Atiq conceived and designed the experiments, performed the experiments, analyzed the data, prepared figures and/or tables, and approved the final draft.
- Amna Kahtan Khalaf analyzed the data, authored or reviewed drafts of the article, and approved the final draft.
- Muhammad Haroon U. Rashid performed the experiments, analyzed the data, authored or reviewed drafts of the article, and approved the final draft.
- Muhammad Asif conceived and designed the experiments, performed the experiments, analyzed the data, prepared figures and/or tables, and approved the final draft.
- Salman Ahmed performed the experiments, authored or reviewed drafts of the article, and approved the final draft.
- Zoha Adil performed the experiments, analyzed the data, authored or reviewed drafts of the article, and approved the final draft.
- Amna Fayyaz analyzed the data, authored or reviewed drafts of the article, and approved the final draft.
- Mohammad Khalid Al-Sadoon analyzed the data, authored or reviewed drafts of the article, and approved the final draft.
- Hamad S. Al-Otaibi analyzed the data, authored or reviewed drafts of the article, and approved the final draft.

## Data Availability

The raw measurements are available in the Supplemental File.

## Supplemental Information

Supplemental information for this article can be found online at http://dx.doi.org/10.7717/peerj.17022#supplemental-information.

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
