# Peer review of "Epidemiology and management of Fusarium wilt of Eucalyptus camaldulensis through systemic acquired resistance"

_PeerJ, doi:10.7717/peerj.17022_

## Round 0.1 · original submission · Major Revisions

Please incorporate the all comments of reviewers and submit a revision along with a point-to-point rebuttal letter. Please also improve the language of the paper.

**Language Note:** The Academic Editor has identified that the English language must be improved. PeerJ can provide language editing services - please contact us at copyediting@peerj.com for pricing (be sure to provide your manuscript number and title). Alternatively, you should make your own arrangements to improve the language quality and provide details in your response letter. – PeerJ Staff

·

Basic reporting

The manuscript titled "Epidemiology and Management of Fusarium Wilt of Eucalyptus camaldulensis Through Systemic Acquired Resistance" holds significant practical relevance by elucidating the impact of plant activators and nutrients on Fusarium wilt of Eucalyptus, within the context of epidemiological factors. The study aligns effectively with the principles of sustainability.
In order to enhance the manuscript prior to publication, the following major revision are proposed:
Line 46: bring up the importance of Eucalyptus camaldulensis
Line 47: write formulae of treatments in proper scientific way
Line 55: use and to separate before the last variable
Line 57: provide clarification regarding the specific disease or range of pathogens that the proposed treatment aims to manage.
Line 58: underscore the future directions are depending on recent exploration
The research title is centered on the exploration of Systemic Acquired Resistance (SAR). In order to effectively convey the essence and significance of SAR within the abstract, it is imperative to articulate and emphasize this phenomenon using precise and specialized terminology.
Line 59: It is imperative to select keywords that diverge from those already incorporated in study title
Line 89: “but Fusarium oxysporum is responsible for Fusarium wilt disease” it would be better to use it like; “but Fusarium Oxysporium responsible for Fusarium wilt of Eucalyptus poses significant threat”
Line 92: use disease name like Fusarium wilt of Eucalyptus for clear understanding of reader
Line 97,98: refrain from duplicating or reproducing the provided information
Line 141: write the autoclave process of your PDA media
Line 142: The stipulated incubation temperature range of "±25°C" for the successful cultivation of the pathogen is challenging to achieve in standard incubators. Write the practically applicable temperature range.
Line 149: from where the seedlings of E. camaldulensis were collected?
Line 150: soil used for pathogenicity test was sterilized or not?
Fluency is lacking in the paragraph of pathogenicity test
Line 165: write the inoculation method you followed under glasshouse
Line 172: write the exact name of your disease
Line 174, 200: it would be better to write the formula like; “ Disease incidence (%) = [(No. of infected plants) / (No. of total plants)] × 100 ” for proper understanding to readers
Line 229: probability is usually considered 5% or p = 0.05, not 0.05%
Line 243-249: reconsider the line for grammatical error
Line 279, 283: use between instead of b/w
Line 301: support your findings “Evaluation of nutrition towards incidence of fusarium wilt, number of leaves along with fresh and dry weight” with at least three relative studies
Line 399: change de-sign to design
Line 410: write future directions for growers based on your revelations.
Some grammatical errors are needed to be corrected

Experimental design

no

Validity of the findings

no

Additional comments

no

Reviewer 2 ·

Basic reporting

This paper will need a minor review in some sentences and the addition of more references as indicated in the attached file.The papers is overall well structured. However, the introduction needs a significant improvement in the last paragraph.

Experimental design

Material and Methods are overall well described. Some sentences require attention as indicated in the attached file.

Validity of the findings

This study was interesting to read. The results and Discussions were overall well written. Some details will need attention as indicated in the attached file.

Additional comments

Overall, this is a nice paper that adds specific information about the climatic factors that maximize the incidence of E. camaldulensis fusarium wilt. The findings are valid and may have important implications for forest ecosystems. My major comment is that the quality of the introduction is inferior compared to the rest of the paper. The introduction does not clearly describe why the research is being conducted, their research hypothesis, and what the authors hope to achieve with this research. A sentence including this critical information should be placed at the end of the introduction.

Annotated reviews are not available for download in order to protect the identity of reviewers who chose to remain anonymous.

---

## Round 0.2 · Minor Revisions

Please incorporate the following comments and submit with point-to-point rebuttal letter:

There are a number of items that need to be corrected before publication.

1 There are several experiments described in the methods that I do not see data for. For example, greenhouse and field experiments are described but only field experiments are presented.

An experiment that tests the different plant activators in combination with the different nutrient treatments is described but not presented (this seems like a very interesting experiment).

2 The methods described that weekly evaluations of disease incidence were made. Which week is analyzed/presented? When were fresh weight and number of leaves measured?

3 line 167, period needed after "management".

4 line 194 Benoic acid has two oxygens, not 3

5 line 265, change "24.567, 16,13" to "24.567, 16, 13%" (missing " " and "%").

6 Throughout, put a space after each comma when the comma is separating different numbers (rather than acting as a hundreds, thousands, millions separator).

7 Figure legends need more information generally. Figures 2 and 4: since three different concentrations of the compound were used, you need to specify which one you are showing here.

8 Figure 3: legend is cutoff. Also do "1, 2, 3" indicate the different concentrations of the treatments? Please indicate what these concentrations actually are either in the figure legend or on the figure itself. I know the concentrations are given in the text, but we don't know in what order they are given in the figure.

9 Figure 5 legend is cut off

10 Figure 4 and 5, please use the names of the acids rather than their chemical formulas (or use both). Most readers won't know the formulas for SA and BA. OK to leave the formula for the potassium compounds.

11 line 57 and 409 typo: "week" > "weak"

Figure 7. Why are there two lines in each panel? Is one the line drawn between data points and the other some kind of local regression that is extrapolating the curve? The line for rainfall (bottom left) where disease incidence goes way up with no data points (~1.5 mm rainfall) seems unsupported. The second line needs to be explained in the legend and corrected for rainfall. Or just remove it and only show the actual line that connects the data points.

·

Basic reporting

Revised paper is accepted for publication

Experimental design

no

Validity of the findings

no

Additional comments

no

---

## Round 0.3 · accepted · Accept

The comments are duly incorporated. The paper is accepted for publication.